# Model for Integrating the Electricity Cost Consumption and Power Demand into Aggregate Production Planning

Camila Matos [1,2], Antônio Vanderley Herrero Sola [1], Gustavo de Souza Matias [2,3], Fernando Henrique Lermen [4,*], José Luis Duarte Ribeiro [5] and Hugo Valadares Siqueira [1]

1   Graduate Program in Industrial Engineering, Universidade Tecnológica Federal do Paraná, Street Doutor Washington Subtil Chueire 330, Ponta Grossa 84017-220, Brazil; matoscamila@hotmail.com (C.M.); sola@utfpr.edu.br (A.V.H.S.); hugosiqueira@utfpr.edu.br (H.V.S.)
2   Industrial Engineering Department, Universidade Estadual do Paraná, Street Comendador Correia Júnior, 117, Paranaguá 83203-560, Brazil; gusmatias@gmail.com
3   Chemical Engineering Graduate Program, Chemical Engineering Department, Universidade Estadual de Maringá, Avenue Colombo 5790, Maringá 87020-900, Brazil
4   Industrial Engineering Department, Universidad Tecnológica del Perú, Avenue Arequipa 265, Lima 15046, Peru
5   Industrial Engineering Department, Federal University of Rio Grande do Sul, Avenue Osvaldo Aranha, 99, Porto Alegre 90035-190, Brazil; ribeiro@producao.ufrgs.br
*   Correspondence: fernando-lermen@hotmail.com; Tel.: +55-(51)-98494-0121

**Abstract:** The constant increases in electricity tax costs and the mandatory contracting of power demand in advance by companies connected to the high-voltage electrical system drive organizations to improve energy planning in their production processes. In addition, market uncertainties make only stochastic methods insufficient for forecasting future production demand. To fill this gap, this study proposes a model that integrates the cost with electricity consumption and power demand into the aggregate production planning, considering the market uncertainties. The model was empirically applied in the food industry, considering a family of potato chips products. From the collected data, a demand forecast was carried out for a later realization of the aggregate planning, using the Holt–Winters forecast model. Before modeling, the new energy demand was calculated, and finally, the model solution verification was performed. In the case study, after application, it was possible to reduce two workers and a cost reduction of R$ 14,288.00. Moreover, the proposal managed to define a power demand that minimized the costs of electric energy and the total costs of the aggregate production planning.

**Keywords:** planning and production control; aggregate production planning; electricity; power demand; production engineering

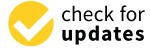



## 1. Introduction

The industrial sector is the largest energy consumer, representing more than a third of primary energy consumption worldwide [1]. With the increasing electricity costs in the recent years, it has become crucial to consider them during production planning [2]. From the producer's point of view, reducing the production process cost can make the product more competitive in the market. Furthermore, within a social context, reducing electricity costs in the productive sector makes it possible to offer more accessible products to the population.

Moreover, the increase in electricity consumption in developing countries follows the population growth and economic progress. In general, it is observed that the population needs to increase the energy use to sustain a better quality of life [3]. Thus, the increase in electricity consumption is one of the results of the economic recovery of emerging countries, but scientists also point it out as one of the factors with the greatest climatic

and environmental impact [4]. It is estimated that if consumption in developing countries continues to follow the current trend, it may surpass that of developed countries [5].

The current challenge for developing countries is to ensure more efficient and sustainable resource management to minimize climate and environmental impacts related to energy production [6]. Unfortunately, Brazil's energy consumption data indicate that the country follows the same line as other developing countries, with increasing consumption over time.

Since the Brazilian energy matrix is heavily dependent on water resources, the low rainfall due to the seasons can become a bottleneck for the Brazilian society and its production chain. In the country, some seasons are systematically characterized by a low amount of rainfall, which influences water resources [5,7]. Given this, decision-makers in Brazil must know how to manage energy consumption better, reducing dependence on thermoelectric plants since they burn fossil fuel, pollute, and have higher generated energy costs, unlike hydroelectric plants.

In this scenario, it is essential that companies efficiently manage their energy consumption, minimize the impact that bad weather has on energy supply, guarantee the efficiency of their operations, and optimize their electricity costs. The efficient use of electrical energy is a way for companies to reduce operating costs and increase their economic performance [8,9].

Several studies in the literature present predictions about energy consumption. Chou et al. [10] employed Machine Learning models and Artificial Neural Networks to predict the demand and ensure the supply to customers of energy supply networks. Hu et al. [8] used prediction models to predict energy consumption; the application case was implemented in energy pipelines to meet the needs and optimize the management. Maaouane et al. [11] used regression models to predict industrial energy demand in Morocco. Mourtzis et al. [12] developed a model to manage energy demand and predicted prices in a way that allows companies to reduce energy costs. On the other hand, no works are found in the literature which differentiates demand forecasting and aggregate production planning.

Aggregate production planning is a way of managing production capacity. Usually, it is carried out considering a period of 2 to 18 months to meet the fluctuation in demand [13]. The aim is to determine the optimal amount of raw material, production, workforce, and inventory levels for each planning period considering limited resource capacities [14,15]. Given the identified gap, this study proposes and applies a model that integrates the cost of electricity consumption and power demand to aggregate production planning. The proposed model allows production planning to consider medium-term energy demand forecasts to reduce waste in procurement in a food production company in Brazil.

This study can assist in the energy planning of companies from different industrial segments since ANEEL (National Electric Energy Agency) Resolution 414/2010 determines that consumers of high voltage electricity must previously contract a power demand to be used throughout the contract period [16]. This requires companies to plan production to meet, as accurately as possible, the amount of demand needed.

To meet the demands of companies, this study aims to answer the research question: how can electric energy and power demand be considered for cost reduction in aggregate production planning? To fill this question, this study aims to propose a model that integrates the cost with electricity consumption and power demand into the aggregate production planning, considering the market uncertainties.

## 2. Theoretical Background

Traditional PAP problem models can be classified into six categories: (i) linear programming (LP); (ii) linear decision rule (RDL); (iii) method of transport; (iv) management coefficient approach; (v) search decision rule (RDB); and (vi) simulation [17]. Feng et al. [18] complemented with three more methods: (i) dynamic programming; (ii) transport tables, and (iii) fuzzy logic mathematical programming. When using any of the planning models, the objectives and inputs of the model are generally considered to be deterministic, and

problems can only be solved if they have the sole objective of minimizing the cost during the planning period [15].

According to Arruda Junior [19], studies have been using models to support aggregate planning in natural systems since the 1980s. For example, Mehdizadeh et al. [20] developed a bi-objective optimization model for an aggregate planning problem with work learning effect and machine deterioration. The first objective maximizes profit, and the second minimizes costs associated with machine repairs and deterioration. The objective was to obtain the appropriate production rates at regular and overtime levels, inventory, and shortages, hiring and firing of workers, and the quantities of subcontractors. To validate the mathematical formulation of the model, it was converted into a single-objective model using the fuzzy objective programming method, based on which computational experiments are performed on a set of small-sized random instances solved by the software LINGO.

Izadpanahi and Modarres [21] performed a linear programming model with three objective functions, which minimize operational cost, energy cost, and carbon emission. It is assumed that several types of energy can be used. They differ in the amount of pollution, calorific value, and cost. The model was applied to the foundry factory, which produced a wide range of products such as aluminum, copper, and lead, and used different machines and manufacturing equipment capable of consuming different types of fossil fuels with different carbon emissions. The objective was to determine the most effective means of satisfying anticipated aggregate demand by adjusting production rates, inventory levels, backlogs, and other controllable variables. The model was solved as a single-objective model, applying a goal-setting technique, and to overcome the effects of uncertain input data, a robust optimization approach was applied to the model. These objective functions sought to minimize operating costs, energy, and carbon footprints. The authors concluded that by increasing the budget level of uncertainty, the amount of increase for electricity cost is much greater than other objective function costs.

Latifoglu et al. [22] presented an optimization model for aggregate production planning under the assumption that electricity supply is subject to uncertain interruptions caused by participation in uninterruptible load contracts. The objective was to minimize the cost of electricity used in production by providing a robust production plan to guarantee demand satisfaction in all possible outage scenarios. The model considered the operational level problem, aggregate production, and inventory planning with the electricity supply uncertainty and deterministic demand. Production decisions were separated from interruption decisions, as the former belonged to the industrial company, while the latter related to the electricity retailer. A heuristic was developed to find a viable solution to the model, in which it found an optimal solution for each instance in 7 periods. The outage uncertainty framework allows different contract and operational rules to be incorporated into the production planning problem simultaneously, such as limiting production in post-outage recovery or prohibiting production level increases in specific periods.

Zhang et al. [2] proposed a new mathematical model to determine efficient scheduling to minimize energy and electricity consumption for the flexible workshop scheduling problem in a time-of-use policy. In addition, a speed selection option has been added to the model, which represents the selection of variable operating speeds. It can be applied to industry and help decision-makers develop production schedules that can reduce time and electricity costs.

Choi and Xirouchakis [23] proposed a definition with a holistic approach to production planning in a reconfiguration manufacturing system with energy consumption and environmental effects. A methodology was defined based on reconfigurable manufacturing. The considered model was developed in linear programming with multiobjective functions to minimize energy consumption and maximize throughput, subject to linear constraints related to various resources and constraints for customer demands. The results showed the developed approach's efficient and practical applicability.

## 3. Materials and Methods

Figure 1 presents the model proposed in this work to integrate energy demand forecasting and aggregate planning, which is composed of four steps: (i) Data collection; (ii) Power Demand Definition; (iii) Production Demand Forecast; and (iv) Aggregate Planning Modeling and Solution.

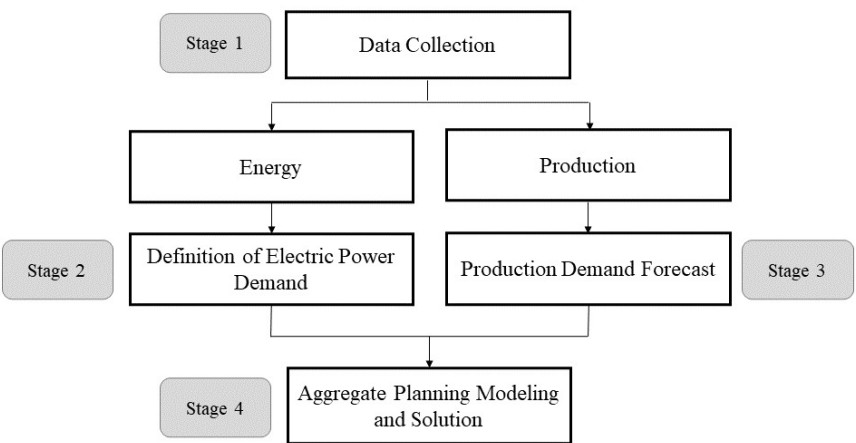

**Figure 1.** Stages of the proposed model.

The development and application of the model were based on a study carried out in a cooperative in the agro-industrial sector, located in southern Brazil, which produces a family of products, namely: straw potato; wavy potato chips; and plain potato chips (traditional and bacon with cheddar). In addition, the cooperative produces an average of 90 tons/month of French fries in a make-to-order environment.

### 3.1. Stage 1—Data Collection

Initially, the product family was identified based on data provided by the studied cooperative. Regarding electricity, it is necessary to collect: the type of contract (blue or green modes); the consumption and demand of power on and off the peak; the energy consumption taxes (R$/kWh), and the power demand (R$/kW) on and off-peak; and the demand for contracted and consumed power.

The peak time, defined by ANEEL—National Electric Energy Agency—in its Resolution 414/2010, occurs between 6 pm and 9 pm when electricity consumption is higher, except on Saturdays, Sundays, and national holidays. The off-peak time is the period composed of the set of consecutive daily hours and complementary to those defined during peak hours.

To discourage the use of electricity at peak hours by high voltage consumers, higher tariffs are imposed at that time. For this reason, the electricity bill has two modalities: blue and green. Consumers powered by 69 kV or more are charged only in the blue mode, while the others can choose blue or green, whichever is more convenient for the consumer. In the blue mode, there is a tariff differentiated by the contracted power demand at the peak and off-peak, and in the green mode, the power demand tariff is unique and can be consumed both at the peak and off [16].

Concerning production, the production demand forecast must be identified; the productive capacity of production; the cost of production; the cost of labor; the cost of hiring employees; the cost of dismissing employees; the forecast of energy consumption; the cost of electricity; the cost of power demand; the number of employees available in the period; and the cost of overtime.

### 3.2. Stage 2—Definition of Electric Power Demand

The charging of the electricity demand taxes, according to Normative Resolution No. 414, of 9 September 2010 [16], is calculated by Equation (1).

$$CDP = TDPC \times DPM \tag{1}$$

where *CDP* is the monthly cost of power demand (R$), *TDPC* is the contracted power demand taxes (R$/kW), and *DPM* is the Measured Power Demand (kW).

Overshoot occurs when an amount above the power demand contracted by the company is consumed. In this way, a different charge is made, obtained from Equation (2) [16].

$$CUDP = (DPM - DPC) \times (2 \times TDPC) \tag{2}$$

*CUDP* is the cost of exceeding power demand; *DPC* is the Contracted Power Demand; *TDPC* is the Contracted Power Demand Taxes.

For example, during the contract period of one year, the company pays monthly for the previously contracted power demand, even if it is not used. Overshoot occurs when the electric power utility measures a value above the power demand contracted by the unit. This tax is stipulated by ANEEL and costs twice as much as contracted. In this condition, the company pays the contracted value plus the value of exceeding demand.

As the company must contract only one demand value charged monthly during the contract period, the proposed model considers each value observed in recent years as a possible power demand to be contracted ($DEC_j$). This is compared with the other measured electrical demands ($DEM_i$) to check for overshoot.

The power demand to be contracted is mandatory and continuously made available by the distributor, at the point of delivery, according to the value and period of validity previously established in the contract, and which must be paid in full, whether or not used during the billing period, expressed in kilowatts (kW). During the entire contract period, for example, one year, the company pays monthly for the previously contracted power demand, even if it is not used. Overshoot occurs when the electric power utility measures a value above the power demand contracted by the unit. The rate for exceeding demand stipulated by ANEEL is double the contracted rate. In this condition, the company pays the contracted value plus the value of exceeding demand. In practice, companies usually contract the highest power demand value measured between the months of the previous year to avoid exceeding demand during the next contract period.

A contracted tax ($TC_i$) is applied to the contracted demand; if consumption is greater than the contracted demand, an overtaking tax ($TU_i$) is also applied. $TC_i$ and $TU_i$ are defined by ANEEL [17]. The cost during the period is the sum of the costs of the '*n*' months of the contract. The power demand to be contracted, then, is the one that results in the lowest cost of the period to be contracted ($\min CDE_j$), obtained by Equation (3).

$$CDP_j = \sum_{i=1}^{n} \left( \left( DPC_j \times TDPC_i \right) + k \times \left( DPM_i - DPC_j \right) \cdot (2 \times TDPC_i) \right) \tag{3}$$

where

$$k = \begin{cases} 1, & DPM_i > DPC_j \\ 0, & otherwise \end{cases}$$

### 3.3. Stage 3—Production Demand Forecasting

This step consists of performing the production demand forecast. The Holt–Winters model was used because it has many uses in demands that present variations, seasonality, and trends. It contains triple exponential smoothing without recalculating seasonal factors from scratch [24,25]. The software R Project x64 (version 3.2.4, R Core Team, Vienna, Austria) and Excel developed the models. The R Core Team [26] developed the R Cran software

to read and model the demand data. The TSA, MASS, tseries, and forecast packages were used for the modeling.

The first manipulation step is plotting the correlogram and performing an initial analysis to identify the components of randomness, trend, and seasonality. When the series is random, autocorrelations close to zero are observed. However, there are drops or positive peaks in the values when there is a trend or seasonality. Therefore, if the data show a trend, it is necessary to carry out the differentiation process to remove this component and then repeat the initial analysis of the differentiated series.

This repetition can take place until the trend component is removed. To prove the assumption of the removal of the trend component, it is necessary to carry out a hypothesis test at 0.05 of significance (Augmented Dickkey–Fuller) [27]. This test is based on the following hypotheses: H0: non-stationary process (null hypothesis); H1: stationary process (alternative hypothesis).

The series is broken down into trend and seasonality in the second step, and the graph is displayed for verification. After this procedure, the database is divided into testing and training, in which 87% train (fit) the model and 13% test the model's prediction quality.

The training process involves a series of steps. First, there is the adjustment of the Holt–Winters model, which can be additive or multiplicative, and, after that, an eight-step forward prediction is performed with these two approaches. Next, the forecasted models must be analyzed to choose the best among them. This analysis is performed from the calculation of forecast errors. Finally, with the results in hand, the prediction made by the model that presents the best performance is chosen.

Then, it is necessary to carry out graphical tests of normality and autocorrelation. The most commonly used test is the Shapiro–Wilk test [28]. The model's simplifying hypotheses are H0: residuals are not normal (null hypothesis); H1: residuals are normal (alternative hypothesis). If the *p*-value of the test is less than 5%, the null hypothesis is rejected so that the model meets the normality assumption.

### 3.4. Step 4—Aggregate Planning Modeling and Solution

The model developed to solve the Aggregate Production Planning (APP) consists of a linear programming model. The model equation was elaborated involving two dimensions, labor and financial. The first case involves hiring and subcontracting personnel. As for production, to carry out aggregate planning, it is necessary to collect the data presented in Section 3.1.

It is necessary to define its horizon to prepare the planning model, denoted in this equation by H. For aggregate production planning problems, the horizon usually varies from three to fifteen months ($3 \leq H \leq 15$). Thus, the H can be changed according to the problem. In this study, H is equal to 12 months since the idea is to develop an annual production plan.

After defining the horizon, it is necessary to determine the main variables and parameters that describe the problem. At this stage, the new reduced electric energy cost and power demand are allocated as variables and have their restrictions. With the literature support, it was possible to define those that will be used in the mathematical equation of the proposed model according to the nomenclature table presented in the Nomenclature list.

To obtain the cost of electricity, consider the electricity taxes (TE) in R$/kWh and an energy performance indicator (I) [29] that measures the relationship between energy consumption electricity by the amount produced (kWh/kg). This represents the average value from January monthly indicators 2014 to December 2018. Thus, the cost of electricity (R$) is obtained by: $C_t^{ee} = I \times TE$.

As this study focuses on integrating this power demand to reduce costs, the power demand cost in the period was considered a parameter, and the contracted power demand was considered a decision variable. In this way, from such definitions, it is possible to build the cost minimization function (Equation (4)) [30]:

$$Min \sum_{t=1}^{T}(C_t^u \cdot U_t + C_t^{ee} \cdot U_t + C_t^e \cdot U_t^e + C_t^m \cdot T_t + C_t^c \cdot T_t^c + C_t^D \cdot T_t^D + C_t^{He} \cdot H_t^e + C_t^S \cdot U_t^S + C_t^f \cdot U_t^f + C_t^{de} \cdot D_t^e) \quad (4)$$

This function has the following restrictions, based on the nomenclature presented in the "Nomenclature list", such as the conservation of workforce, the limitation of production in the period, the limits of overtime, the limit of stock, the limits of backorders, the power limitation, and non-negativity constraints.

In the first constraint (Equation (5)), the number of workers in the current period ($T_t$) must be equal to the number of workers in the previous period ($T_{t-1}$) added to the hirings in the current period ($T_t^c$) and subtracted from layoffs in the current period ($T_t^D$), being [30]:

$$T_t = T_{t-1} + T_t^c - T_t^D \text{ para } 1 \leq t \leq H \quad (5)$$

The second constraint (Equation (6)) is characterized by the limitation of production in period $t$, in which the produced capacity cannot exceed the available capacity. This is internally defined based on available labor hours, regular or extra. Thus, it is determined by the amount that the company can produce in normal hours per month ($H_t^N$) and in overtime ($H_t^H$) [30]:

$$Ca_t = H_t^N + H_t^H \text{ para } 1 \leq t \leq H \quad (6)$$

In the third constraint (Equation (7)), there is a limit for overtime (established by the company itself); that is, if it is not desired that overtime production exceeds 25% of production capacity in each period ($H_t$), the constraint appears to be [30]:

$$H_t \leq 0.25 \text{ to } 1 \leq t \leq H \quad (7)$$

This variable is balanced at the end of each period in the stock restriction. The net demand is obtained from the sum of the demand $D_t$ and the number of units missing $U_{f-1}$ from the previous period. Thus, this can be met by current production (in normal $U_t$, subcontracting $U_t^S$, extra hour $U_t^H$ and by previous stock $E_{f-1}$), or part of it is accumulated. This relationship is represented by Equation (8) [30]:

$$U_t^e = E_i + H_t^n + U_t^{He} + U_t^S - D_t - U_{f-1}, \text{ para } 1 \leq t \leq H \quad (8)$$

The backorder restriction (Equation (9)) is limited to the number of products allowed by the company that will be out of stock, being [30]:

$$U_t^f \leq X \text{ to } 1 \leq t \leq T \quad (9)$$

The power limitation constraint (Equation (10)) is limited to contracted demand power [30].

$$D_t^{pe} = X, \text{ to } 1 \leq t \leq H \quad (10)$$

The non-negativity constraints indicate that the number of units produced ($U_t$), in final stock ($U_t^e$), of available workers ($T_t$), hired ($T_t^c$), and fired ($T_t^D$), must be greater than or equal to zero, according to Equation (11) [30].

$$U_t ; U_t^e ; T_t ; T_t^c ; T_t^D; U_t^s; U_t^f; H_t \geq 0 \text{ to } 1 \leq t \leq H \quad (11)$$

All parameters and variables that involve the production process were considered. However, according to the company's characteristics, they can be removed from the equation during the application or reset to zero. Thus, the resulting model determines the most effective means of satisfying the expected aggregate demand, adjusting production rates, the amount of electrical energy, and other controllable variables.

To solve the model, the equation is solved with the help of the Solver tool of the Excel software. Then, as planned, the production plan for the next 12 months was drawn up.

After this procedure, the current scenario of the cooperative is compared with the proposed scenario using the new power demand.

### 3.5. Model Verification (Sensitivity Analysis)

Sensitivity analysis consists of verifying the stability of the solution from possible variations of the parameters used in linear programming; that is, it determines how the solution can be modified from changes in its parameters [31]. For Butler et al. [32], it is the one who determines the robustness of the solutions obtained. In this analysis, some points are verified, in which Bazaraa et al. [33] reported that the shadow price is associated with the constraints of Linear Programming; if it increases, the objective function also changes in the same proportion, such as, for example, each more unit of raw material that is available for production also increases the profit in the objective function.

The reduced cost of variables measures the impact on the objective function caused by the entry of a variable unit in the solution. Thus, when a variable presents positive values, it does not participate in the optimal solution, and a penalty is to be paid if used. Sensitivity analysis was performed using the Solver tool of the Excel Software (version 3, Microsoft, Seatle, WA, USA).

## 4. Results

This section is separated into five subsections: (i) Overview of the collected data; (ii) Definition of New Power Demand; (iii) Production Demand Forecast; (iv) Modeling and Solution of Aggregate Planning in the Cooperative; (v) Production plan for the case studied; and (vi) Model Verification from Sensitivity Analysis. This research is also delimited in the economic sector based on the objectives outlined. It was developed in a company in a financial segment geographically located in the southern region of Brazil. As for the branch of activity, it is agro-industrial. The type of activity fits into a single family of products, with a specific profile to produce French fries.

Due to the cooperative opening and interest in the proposed study, the model was applied in an agro-industrial cooperative in southern Brazil. It comprises 877 members and 3153 employees, with a turnover of 2.91 billion reais, with business units divided into operations (agricultural, meat, milk, potato, beans, and administration) and industrial (meat, milk, potato, and beer). For example, the French fries unit has 36 employees (17 in the potato production line and the rest in administration and other sectors), producing an average of 90 tons/month of French fries in packages ranging from 80 g to 380 g.

This study was carried out in a make-to-order-rated environment. This system is dynamic and has flaws, such as a lack of operators and materials and low adherence to production orders due to pre-established delivery times [34,35].

The cooperative produces only a single family of products: potato straw (traditional, extra-fine, and very fine), wavy potato chips (traditional, meat, cheese and onion, and salsa), and smooth potato chips (traditional and bacon with cheddar). Its production process is represented in Figure 2.

Initially, the raw material arrives at the unit already processed (cleaned and peeled). Afterward, the potatoes are washed and directed by an infinite conveyor for a selection, in which the employees cut the defects presented. Next, they follow the conveyor belt and go through the slicing stage, utilizing a cutter according to the model of the potato produced (straw potatoes, wave potato chips, and plain potato chips). Then, the sliced potatoes are washed to remove the starch.

In the next step, the excess water is removed, and they go on to the oil frying step by submersion using mats. Afterward, the potatoes are cooled and seasoned, and a selection is made to remove those outside standards. Finally, the ready-made potatoes go to packaging, filling them into the packages together with nitrogen; the packages are placed in cardboard boxes and sent to the shipment as a finished product.

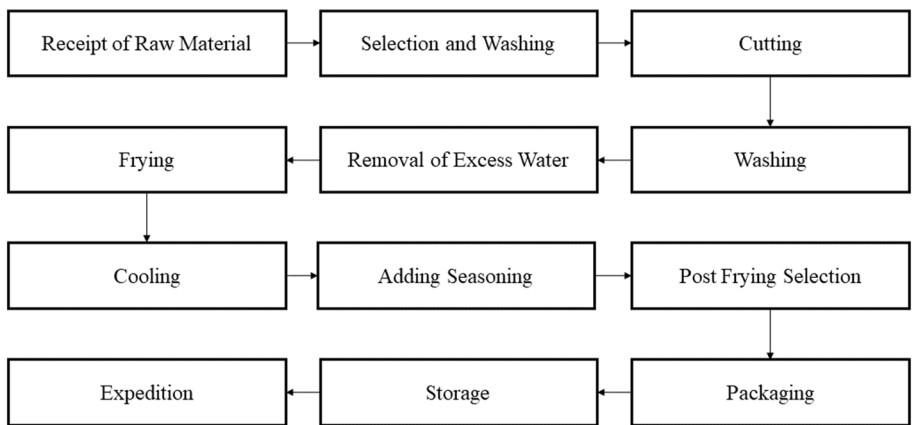

**Figure 2.** Potato production process.

### 4.1. Overview of the Collected Data

The power demand contracted by the industry in the last contract period was 255 kW. The consumer unit pays the exceeding part if more than 5% of the contracted demand is consumed. Because of this, the contracted power demand taxes have a value of R$ 13.14/kW, and the overtake taxes, R$ 26.28/kW.

From the history of electric power consumption over the last five years, available in Appendix A (Table A1), it was possible to plot a graph to analyze the behavior of this consumption, as shown in Figure 3.

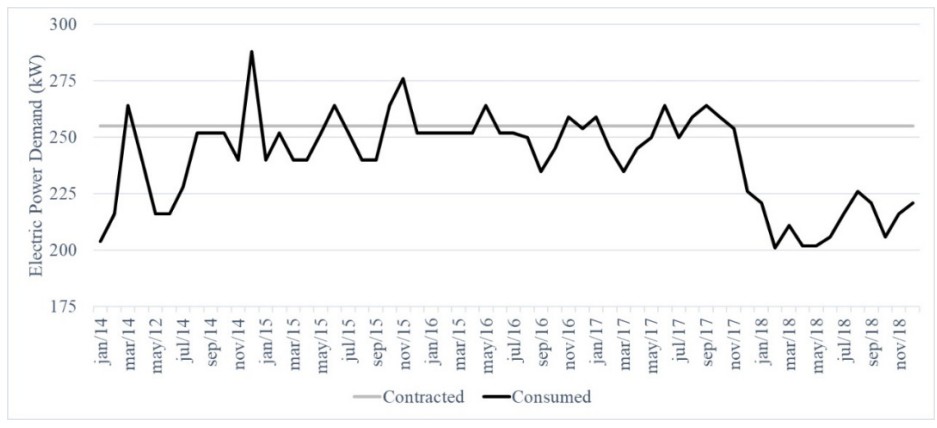

**Figure 3.** Demand for contracted and consumed electrical power over time.

From Figure 3, it is possible to see that, over the last five years, the consumer unit exceeded the contracted power demand in some months, and in most of them, it did not use the contracted value of 255 kW. In 2014 and 2016, consumption was exceeded in just two months, while in the other months, there was a loss of contracted power demand. In 2015, there were three overtaking. It was in 2017 when most overtaking occurred, totaling five months. On the other hand, during 2018, the consumer unit paid for a much higher demand than used. This shows a lack of reconciliation between production planning and energy planning. Two overruns in the contracted demand generated a fine, causing losses to the company, highlighting the need to carry out this study.

### 4.2. Definition of New Power Demand

From the data collected, all the power demands measured from January 2014 to December 2018 were organized in a table (Appendix A, Table A2). From this, and with the help of Equation (3), it was possible to obtain the total costs of each value and, consequently, define the best among them, as shown in Figure 4.

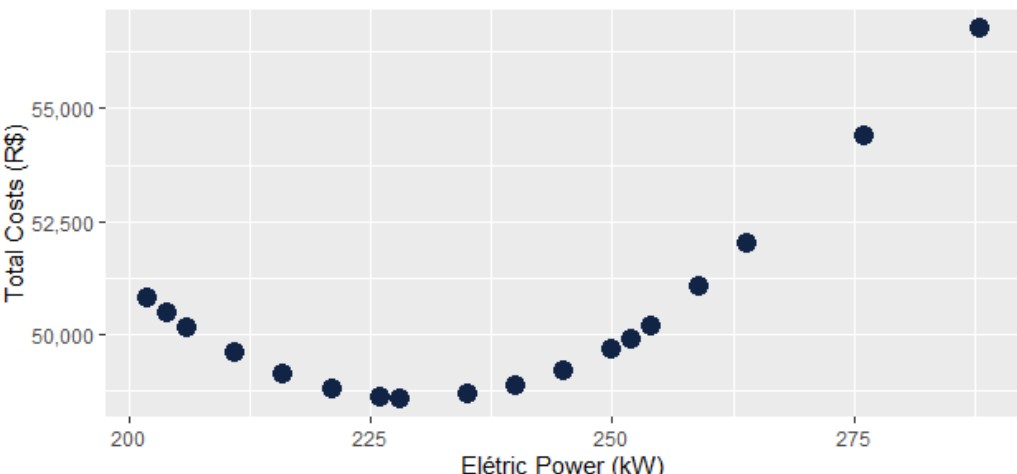

**Figure 4.** Power demand costs.

As shown in Figure 4, it is noted that the 240 W and 235 W demands had the exact value of R$ 67,881.24. Thus, the demand to be contracted by the cooperative is 235 W.

### 4.3. Production Demand Forecast

To perform aggregate planning, it is needed to identify demand forecast data. As the studied company did not have its demand forecast, it was necessary to perform it. Otherwise, only data would be collected.

To prepare the production demand forecast, the historical data of the production of kg of potato were considered in the period of 60 months, from January 2014 to December 2018, given in Appendix A (Table A3). Therefore, the first step was to analyze the data profile surveyed to identify the components present, as shown in Figure 5.

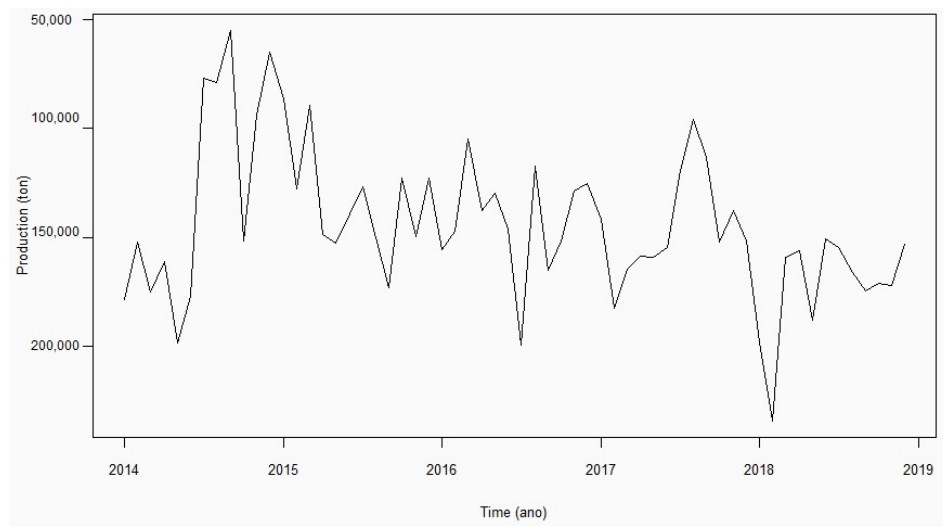

**Figure 5.** Potato production history from January 2014 to December 2018.

From Figure 5, it is possible to notice that potato production is a series that shows a slight downward trend with high and low peaks. However, it is also observed that it does not present a well-defined pattern of seasonality and cyclicality. Therefore, the correlogram was plotted to understand if the series was random or if it had any tendency or seasonality, as seen in Figure 6. When the series is random, it has autocorrelations close to zero. However, when there is a trend or a seasonality, the series shows a downward trend or positive peaks in the values.

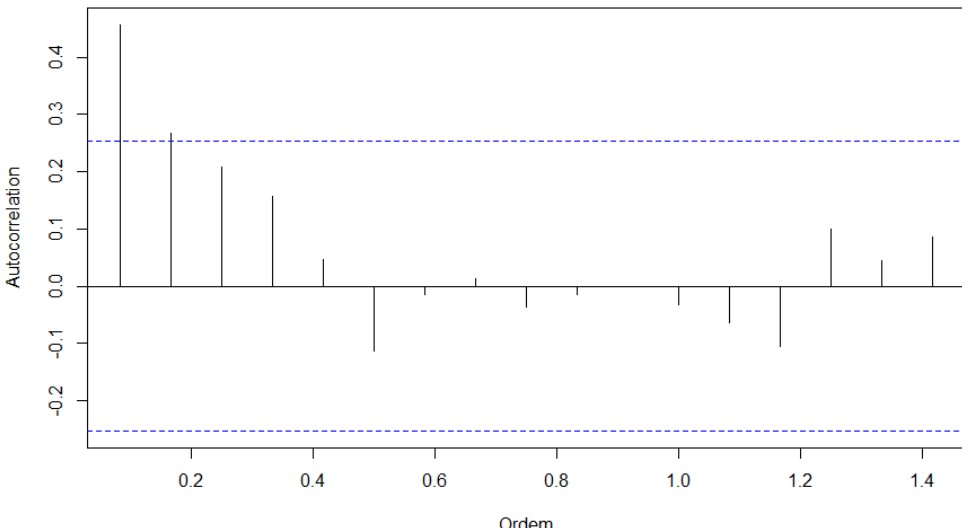

**Figure 6.** Correlogram of the potato production time series with normalized data.

In Figure 6, the vertical axis indicates autocorrelation, and the horizontal axis indicates lag. The blue dashed line indicates where it is significantly different from zero. Note that all values except the first are within the blue dashed line limit. This means zero autocorrelation, indicating that potato production can be random and supposedly stationary. On the other hand, if the values were above the dashed line, the series would not be random and would have to be treated with a moving average, which is not the case. However, to be sure of the statements, it was necessary to carry out a test to prove the assumptions.

The Augmented Dickey–Fuller (ADF) test resulted in $-3.5467$, Lag order = 3, and p-value = 0.0453, confirming the alternative hypothesis that the process is stationary. In this case, potato production can be random and stationary. From the ADF test, the series decomposition into trend and seasonality was performed, as shown in the graph in Figure 7.

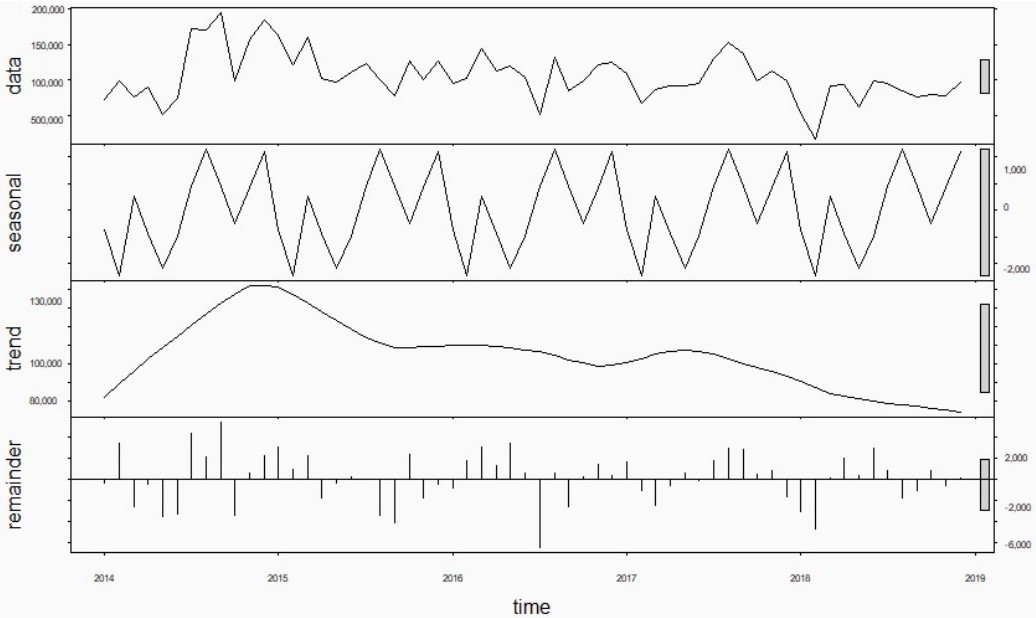

**Figure 7.** Components of trend and seasonality.

After differentiating the actual demand, the components' performance and behavior show seasonality in the production of potatoes, always in the same period over five years. As for the trend, it is possible to observe a slight growth in 2014; from 2015, it started to

decrease. This can be explained by the crisis in mid-2014, with the drop in the Brazilian Gross Domestic Product (GDP) [36], which directly affected the cooperative's sales.

The STL algorithm performs time series smoothing using the LOESS algorithm [37] in two loops; the inner loop iterates between seasonal and trend smoothing, and the outer loop minimizes the effect of outliers. During the inner loop, the seasonal component is calculated first and removed to calculate the trend component. Then, the remainder is calculated by subtracting the seasonal and trend components from the time series. Decomposing seasonal trends using LOESS (STL) is a robust time-series decomposition method often used in economic and environmental analyses. The STL method uses locally adjusted regression models to decompose a time series into trend, seasonal, and remainder components [38–40].

The series was divided into two sets of data, the first consisting of samples from 1 to 52 and the second from 53 to 60, to check the quality and to plot the profile of the additive and multiplicative Holt–Winters models, as shown in Figure 8.

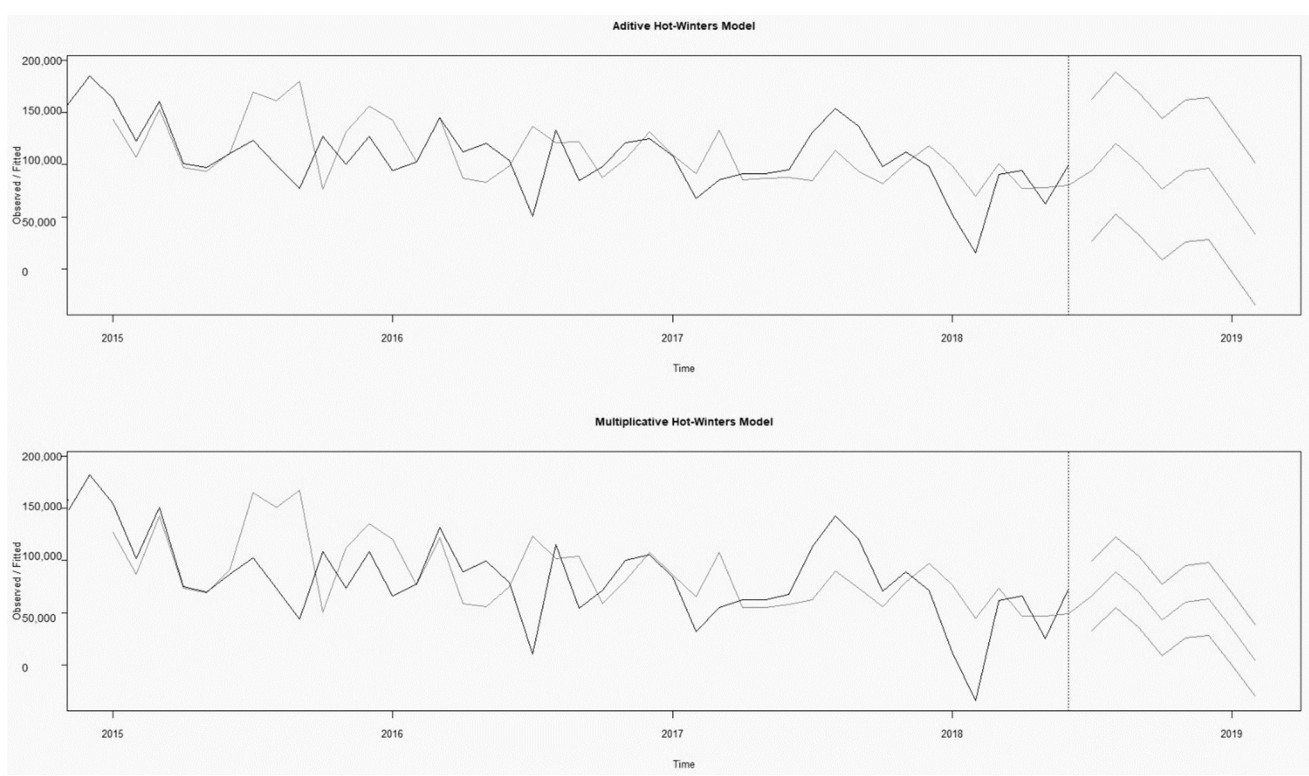

**Figure 8.** Holt–Winters additive and multiplicative model.

It can be noticed from Figure 8 that the profile behavior of the two models is close to the profile of the original data. However, this visual analysis does not define the best model to represent the demand forecast. For this reason, forecast errors were calculated. The calculation results are shown in Table 1, where REQM is the root mean square error, MAE is the mean absolute error, and MAPE is the mean percentage of the absolute error.

**Table 1.** Model performance measures.

| Models | REQM | MAE | MAPE |
|:---:|:---:|:---:|:---:|
| Additive | 18,928.18 | 14,549.97 | 62.30 |
| Multiplicative | 15,047.58 | 13,076.68 | 28.61 |

From Table 2, it is possible to verify that the multiplicative model has the lowest error considering all metrics; that is, its performance is better for forecasting production demand. Therefore, the errors of the multiplicative model were statistically tested in order to identify

normalities. The Shapiro–Wilk test resulted in W = 0.929 and *p*-value = 0.0152, thus rejecting the null hypothesis. This indicates that the multiplicative model is normal. Afterward, the production demand forecast was carried out for the next 12 months, as shown in Table 2.

**Table 2.** Production demand forecast.

| Month | Production Demand Forecast (kg) |
|---|---|
| January | 69,579.00 |
| February | 45,862.00 |
| March | 84,995.00 |
| April | 74,710.46 |
| May | 69,592.76 |
| June | 71,464.08 |
| July | 82,127.11 |
| August | 97,713.42 |
| September | 84,283.04 |
| October | 66,475.00 |
| November | 77,758.62 |
| December | 79,578.84 |

It is noted that the production forecast presents a variation between 45,862.00 kg and 97,713.00 kg, representing 46.9%.

The demand forecast performed will be used in aggregate production planning. Because to do it, it is necessary to know the demand forecast.

*4.4. Modeling and Solution of Aggregate Planning in the Cooperative*

The strategy used to develop aggregate planning monitored demand with constant labor and overtime. This strategy provides continuity of labor and avoids emotional and tangible costs of hiring and firing personnel. The data used for the chosen strategy, as shown in Table 3, were collected with the cooperative understudy and calculated.

**Table 3.** Cooperative Costs.

| Costs | Price (R$) |
|---|---|
| Production unit | 15.00/Kg |
| Labor | 2153.25/month |
| Regular time | 13.45/hour |
| Extra hour | 23.53/hour |
| Hiring | 2100.00/worker |
| Resignation | 18,120.48/worker |
| Electricity taxes | 0.30/R$/kWh |
| Power demand | 13.14/Watts |

The company calculates the unit cost of production, encompassing all costs involved in the process. As for labor, during regular hours, the workers receive a salary of R$ 1485.00, but the real value that the company has with this cost is R$ 2153.25 (R$ 13.45 × 8 h/day × 20 days/month) due to charges (R$ 668.25) paid every month. As for the cost of overtime, 75% of regular hours is added.

As for the hiring and firing costs, the high value of both justifies why the cooperative does not use the hiring and firing strategy. The cost of hiring comprises recruitment,

medical costs (doctor and nurse), registration and integration, and training costs. The dismissal involves even more costs, with a 40% fine on taxes (R$ 640.00), termination (R$ 475.00), prior notice (R$ 1485.00), vacation (R$ 1485.00), 1/3 vacation (R$ 495.00), 13th salary (R$ 1485.00), salary (R$ 1485.00), and charges (R$ 10,570.28). It is worth mentioning that these values are variable, depending on the employee's working time, which is at least one year of work in the cooperative.

To obtain the cost of electricity, the tax value (TE = 0.30 R$/kWh) from the contract with the concessionaire (Table 3) was multiplied by the value of the energy performance index (I), obtained by the average of the last five years. Correlation analysis of electricity consumption as a production function showed a coefficient of determination $R^2 = 0.64$. Some values of atypical months were then excluded (mainly as a result of the economic crisis in the country from 2014 onwards) as they are extreme values or values that are further away from the straight line (outliers), and $R^2 = 0.77$ was obtained. Thus, the average value of the energy performance indicator was I = 0.80 kWh/kg, with a Standard Deviation = 0.126 and error = ±0.017. Thus, the unit cost of electricity was $C_t^{ee} = 0.24$ R$/kg.

The cost of production, electricity, and demand for electricity is proportional to the production volume. In each period, demand must be satisfied, as orders are made to order over time can be used to fulfill the plan. Physical resources are considered fixed during the 12-month planning horizon. The total cost verified during the planning horizon is the sum of all the mentioned costs, given by Equation (4), being presented with the data replaced by Equation (12).

$$Min \sum_{t=1}^{T} (18.37 \times U_t + 0.24 \times U_t + 2153.00 \times T_t + 2100.00 \times T_t^c + 18,120.48 \times T_t^D + 3.14 \times D_t^{pe} + 23.53 \times H_t^e) \quad (12)$$

The study aims to find an aggregate plan that minimizes the total cost incurred during the planning horizon. Therefore, the values are subject to a series of restrictions linked to the decision variables presented in Section 3.4.

As for the workforce restriction, hiring, and firing (Equation (5)), the cooperative does not carry out the hiring and firing strategy, so the number of workers in the current period will always be the same. As for the production limitation constraint (Equation (6)), the capacity is determined by the company's normal hours per month and overtime. According to information provided by the cooperative, the production capacity is 88,000.00 kg per month, and it is possible to produce 550 kg/h or 88,000.00 kg/month at normal hours (8 h/day).

The overtime (Equation (7)) is carried out with all workers on the line. The cooperative works with the minimum number of people necessary to meet demand, with only two overtime hours allowed per day. The cooperative is limited to 40 monthly overtime hours/worker in this case. For the power demand constraint (Equation 10), the power demand value that minimizes the total costs was found, so it must be maintained for all planning periods, 235 W.

### 4.5. Production Plan for the Case Studied

With the tool used in the research work, it was possible to prepare the production plan for the next 12 months and, subsequently, compare the current scenario with the proposed one, considering the new power demand.

### 4.5.1. Current Scenario

The company's current scenario is to keep the workforce constant, using overtime when necessary and contracting 255 watts of power demand (Table 4).

**Table 4.** Aggregate plan of the current scenario with the contracted power demand.

| Period (month) | Contractors (Number of People) | Fired (Number of People) | Workforce (Number of People) | Production in Overtime (Hours) | Production (kg) | Power Demand (kW) | Production Demand (kg) |
|---|---|---|---|---|---|---|---|
| 0 | 0 | 0 | 17 | 0 | 0 | 255 | 0 |
| 1 | 0 | 0 | 17 | 0 | 73,058.02 | 255 | 73,058.02 |
| 2 | 0 | 0 | 17 | 0 | 48,155.40 | 255 | 48,155.40 |
| 3 | 0 | 0 | 17 | 0 | 80,745.51 | 255 | 80,745.51 |
| 4 | 0 | 0 | 17 | 0 | 70,974.94 | 255 | 70,974.94 |
| 5 | 0 | 0 | 17 | 0 | 73,072.40 | 255 | 73,072.40 |
| 6 | 0 | 0 | 17 | 0 | 75,037.28 | 255 | 75,037.28 |
| 7 | 0 | 0 | 17 | 0 | 78,020.75 | 255 | 78,020.75 |
| 8 | 0 | 0 | 17 | 9 | 92,827.75 | 255 | 92,827.75 |
| 9 | 0 | 0 | 17 | 0 | 80,068.89 | 255 | 80,068.89 |
| 10 | 0 | 0 | 17 | 0 | 69,798.75 | 255 | 69,798.75 |
| 11 | 0 | 0 | 17 | 0 | 81,646.55 | 255 | 81,646.55 |
| 12 | 0 | 0 | 17 | 0 | 83,557.78 | 255 | 83,557.78 |
| Total cost | | | | 14,301,554.61 | | | |

It can be seen from Table 4 that the production demand (maximum 88,000.00 kg/month) is met, except in the eighth period (in bold), in which it will be necessary to perform nine overtime hours to meet the production demand.

4.5.2. Innovative Scenario with the Proposed Power Demand

Keeping the strategy used by the company of constant workforce, it was possible to define a new plan with the new defined electric power demand, as shown in Table 5.

**Table 5.** Aggregate plan with the proposed power demand.

| Period (Month) | Contractors (Number of People) | Fired (Number of People) | Workforce (Number of People) | Production in Overtime (Hours) | Production (kg) | Power Demand (kW) | Production Demand (kg) |
|---|---|---|---|---|---|---|---|
| 0 | 0 | 0 | 17 | 0 | 0 | 235 | 0 |
| 1 | 0 | 0 | 17 | 0 | 73,058.02 | 235 | 73,058.02 |
| 2 | 0 | 0 | 17 | 0 | 48,155.40 | 235 | 48,155.40 |
| 3 | 0 | 0 | 17 | 0 | 80,745.51 | 235 | 80,745.51 |
| 4 | 0 | 0 | 17 | 0 | 70,974.94 | 235 | 70,974.94 |
| 5 | 0 | 0 | 17 | 0 | 73,072.40 | 235 | 73,072.40 |
| 6 | 0 | 0 | 17 | 0 | 75,037.28 | 235 | 75,037.28 |
| 7 | 0 | 0 | 17 | 0 | 78,020.75 | 235 | 78,020.75 |
| 8 | 0 | 0 | 17 | 9 | 92,827.75 | 235 | 92,827.75 |
| 9 | 0 | 0 | 17 | 0 | 80,068.89 | 235 | 80,068.89 |
| 10 | 0 | 0 | 17 | 0 | 69,798.75 | 235 | 69,798.75 |
| 11 | 0 | 0 | 17 | 0 | 81,646.55 | 235 | 81,646.55 |
| 12 | 0 | 0 | 17 | 0 | 83,557.78 | 235 | 83,557.78 |
| Total cost | | | | 14,298,401.01 | | | |

Keeping the strategy, what differentiates this plan (Table 5) from the previous one (Table 4) is the reduced cost of R$ 3153.60 reais due to the new power demand contracted for this year. However, even with this reduction, another plan was prepared to minimize total costs if the company allowed the hiring and firing of employees, if necessary. The result is shown in Table 6.

**Table 6.** Aggregate plan as a proposed scenario considering power demand.

| Period (Month) | Contractors (Number of People) | Fired (Number of People) | Workforce (Number of People) | Production in Overtime (Hours) | Production (kg) | Power Demand (kW) | Production Demand (kg) |
|---|---|---|---|---|---|---|---|
| 0 | 0 | 0 | 17 | 0 | 0 | 235 | 0 |
| 1 | 0 | 2 | 15 | 0 | 73,058.02 | 235 | 73,058.02 |
| 2 | 0 | 0 | 15 | 0 | 48,155.40 | 235 | 48,155.40 |
| 3 | 0 | 0 | 15 | 0 | 80,745.51 | 235 | 80,745.51 |
| 4 | 0 | 0 | 15 | 0 | 70,974.94 | 235 | 70,974.94 |
| 5 | 0 | 0 | 15 | 0 | 73,072.40 | 235 | 73,072.40 |
| 6 | 0 | 0 | 15 | 0 | 75,037.28 | 235 | 75,037.28 |
| 7 | 0 | 0 | 15 | 1 | 78,020.75 | 235 | 78,020.75 |
| 8 | 0 | 0 | 15 | 28 | 92,827.75 | 235 | 92,827.75 |
| 9 | 0 | 0 | 15 | 4 | 80,068.89 | 235 | 80,068.89 |
| 10 | 0 | 0 | 15 | 0 | 69,798.75 | 235 | 69,798.75 |
| 11 | 0 | 0 | 15 | 7 | 81,646.55 | 235 | 81,646.55 |
| 12 | 0 | 0 | 15 | 11 | 83,557.78 | 235 | 83,557.78 |
| Total cost | | | | 14,284,113.02 | | | |

Note that this plan (Table 6) allows for dismissal and hiring and presents a lower cost for the cooperative than the previous plan (Table 5). However, it is worth mentioning that it was only allowed to fire two people because the production line could not operate with less than 15 workers. Thus, in the months in which demand was not satisfied, only with the staff, it would be necessary to work overtime, as shown in Table 6. In this way, there would be a cost reduction of R$ 14,288.00.

*4.6. Model Verification from Sensitivity Analysis*

The sensitivity analysis is divided into sensitivity analysis on variable cells and sensitivity analysis on constraints; the results presented were collected from the report issued by Solver in the resolution of the aggregate planning.

The first scenario, with a constant workforce and new power demand, shows that the reduced cost of the variables is either positive or equal to zero; that is, the variables are part of the solution, and there is no slack. The exception is the dismissed workers variable, which has a negative reduced cost; it shows that it has extra workers for production due to the cooperative's capacity. As for the permission to increase or reduce, without modifying the optimal solution, the only variable that allows an increase in its cost is that of the workforce, presenting a different increase in each period, and the variables that enable the reduction are the production of hours overtime, production, and laid-off workers. The one for dismissed workers allows a different value for each period, while the others present the same value for all. Variables with a value equal to zero cannot be changed because the optimal solution is also changed.

As for restrictions, contractors, production, and overtime have a favorable shadow price; the objective function will increase when they are increased. For example, if the number of overtime hours increases, the cost of the objective function will also increase. The workforce, in turn, has a negative shadow price; if the workforce increases, the optimal

solution is not changed since the Solver interprets the surplus of workers due to production capacity. This is because 17 workers have been fixed as a restriction of the cooperative. As for the permission to increase, it is possible to increase the cost of the workforce by up to R$ 0.85 in all periods, those hired by up to R$ 0.93 in the first eight periods, overtime by 671.22 h in the eighth period, and in all others in 680 h, the maximum limit that can be accomplished. The production constraint has a different increase limit for each period. As for the permission to reduce, the restriction of contractors does not allow reduction due to the limit set by the cooperative. The workforce can be reduced by 0.93 kg in the first eight periods, while it cannot be reduced in the others. Production can only be reduced in the eighth period by 4827.76 kg.

In the second scenario, allowing hiring and firing with the new power demand in the variables, the reduced cost remains the same as in the previous scenario. It can be noted that the dismissed workers, in this scenario, have zero cost, different from the last scenario, because now the restriction allowed to keep at least 15 workers, no more than necessary. As for the permission to increase or reduce, without modifying the optimal solution, the only variable that allows an increase in its cost is laid-off workers, as two of them are laid off at the beginning of the period. In the other periods, it is allowed to increase the cost in the variable workforce in all and the variable overtime in periods 3, 7, 8, 9, 11, and 12. All variables allow a reduction limit.

As for the restrictions and the workforce continue to have a negative price shadow in the first nine periods and the 12th period, the Solver continues to consider production less than capacity concerning workers. However, according to the cooperative manager, 15 workers present the minimum framework for the show. On the other hand, periods 10 and 11 have a positive price shadow; if there is an increase in the workforce in this period, the optimal solution may change. The other constraints also have a positive or zero value.

## 5. Discussion

The planning horizon defined for this study was 12 months. Entezaminia et al. [41] stated that it is usually performed from 3 to 18 months. Filho et al. [42], and Feng et al. [19] also planned with a horizon of 12 months. On the other hand, Kopanos et al. [43] used a horizon of only five days and, Bilgen and Dogan [44] of 7 days and three weeks.

The formulated model addressed the demand monitoring strategy, using overtime, hiring, and firing. The strategy depends on the reality of the place, as is the case of studies that considered the stock (i.e., [27–31]), subcontracting [25,29,32], and missing orders [21].

This work is similar to that of Mehdizadeh et al. [21] regarding aggregate planning since it determined the number of items to be produced in regular and overtime and the need to keep the workforce constant or not, as well as Kopanos et al. [43] and Bilgen and Dogan [44]. They also determined a regular production plan. In contrast, Feng et al. [19] used APP also to obtain the production lead time.

This presented model was formulated with linear programming, following Rajaram and Karmarkar [45], who stated that APP models are generally designed as linear programming that minimizes costs. The model formulated in this study sought to minimize production costs as demand was met. Therefore, it is possible to affirm that, in the literature discussed here, the planning models usually seek to minimize costs through strategies according to the reality of the study site. For example, Filho et al. [42] considered inventory levels, production rates, overtime, regular labor, and subcontracting. On the other hand, Feng et al. [18] used minimum inventory costs, ideal production rate, and production balance load rate, seeking minimization.

Kopanos et al. [43] focused on minimizing inventory, operating, batch recipe preparation, unit usage, family moving, and external production costs. On the other hand, Bilgen and Dogan [44] sought to maximize their production, as did Kadambur and Kotecha [46], who also aimed to maximize profit while determining the amount and type of product that should be produced. In this way, these authors go in the opposite direction to Wang

and Liang [15], who claimed that the problems of any of the planning models can only be solved if they have the sole objective of minimizing the cost.

A similar work to this study is that of Izadpanahi and Modarres [21], which included energy cost and carbon emission in the model. The authors used three objective functions seeking to minimize such costs simultaneously. The main point was to find the ideal power demand value before including it in the aggregate planning. This contributes to the literature since even some studies considering electric energy in their costs do not consider the demand for power and, consequently, did not seek to reduce it before adding it. Mehdizadeh et al. [20] reported that aggregate planning costs involve inventory, ordering, and production costs, and yet, Stevenson [47] complemented with unit costs related to regular hours, overtime, subcontracting, layoffs, or other factors that affect the costs in a relevant way, that is, it does not consider the cost of electricity.

Choi and Xirouchakis [1] minimized energy consumption while maximizing production based on a multiobjective function. Mouzon et al. [48] sought to reduce the energy used by production equipment based on mathematical programming, similar to the study by Rager et al. [49], who reduced energy use in production by minimizing the final energy demand of the machines involved in the process. Finally, Santiago et al. [50] and Chaturvedi [31] aimed to reduce energy in the process facilities and shared resources of aggregate planning.

In contrast to these studies, this paper sought to include the power demand in the planning, considering minimizing its cost from an equation before adding it to the other existing ones. By having the cost of electricity in the aggregate planning, the proposed model contributes to the literature and companies connected to the high-voltage electrical system that need to contract power demand in advance. It is essential to highlight that the models found in the literature present a production schedule based on stochastic data.

## 6. Conclusions

This study aimed to propose an aggregate planning model integrating the cost with electrical energy and power demand. The model is innovative in three aspects: the cost of contracted power demand in aggregate planning, defining the power demand to be contracted based on the least cost principle, and the treatment of uncertainties in the production demand forecast with the participation of the decision-maker.

The application of the model was carried out in a cooperative of the small agro-industrial sector. However, this study provides insights for a company of any size that needs to contract power demand in advance and presents a high degree of uncertainty in the market. After application, it was possible to reduce two workers and a cost reduction of R\$ 14,288.00. The main limitation of the proposed model is the single case study. Furthermore, although it has been applied in a make-to-order business, it can also be used in make-to-stock or project-to-order businesses.

A characteristic of the company used in the case study is that customers usually confirm their orders a few days before shipment; such orders can be changed. Thus, the suggestion for future study would be to generate a production plan at the end of the month, but as the power demand must be contracted in advance for an extended period, this can be considered impossible. Thus, for these cases, the suggestion is that the power demand forecast is revised, if the contract allows, to reduce the period and generate a plan with a smaller number of months, or even to integrate the power demand equation in the resolution of the aggregated planning so that, when solving it, the best option is presented.

**Author Contributions:** Conceptualization, C.M.; Data curation, G.d.S.M. and C.M.; Formal analysis, G.d.S.M. and C.M.; Investigation, C.M.; Methodology, C.M. and F.H.L.; Project administration, A.V.H.S.; Resources, C.M.; Supervision, A.V.H.S.; Writing—original draft, C.M.; Writing—review & editing, A.V.H.S., F.H.L., J.L.D.R. and H.V.S. All authors have read and agreed to the published version of the manuscript.

**Funding:** This research was funded by CAPES Foundation and The APC was funded by Universidad Tecnológica de Peru.

**Institutional Review Board Statement:** Not applicable.

**Informed Consent Statement:** Not applicable.

**Data Availability Statement:** The data will be provided by the corresponding author if asked.

**Conflicts of Interest:** The authors declare that they have no known competing financial interests or personal relationships that could have appeared to influence the work reported in this paper.

### Nomenclature

| | |
|---|---|
| $(DEC_j)$ | **power to be contracted** |
| $(DEM_i)$ | **Other measured demands** |
| $(TC_i)$ | **Contracted taxes** |
| $(TU_i)$ | **Overtaking taxes** |
| **PARAMETERS** | |
| H | Planning horizon in periods |
| t | Period index |
| $D_t$ | Expected demand for period $t$ |
| $C_t^u$ | Unit cost of production in period $t$ |
| $C_t^{ee}$ | Unit cost of electricity in period $t$ |
| $C_t^{de}$ | Power demand cost in period $t$ |
| $C_t^e$ | Inventory cost of one unit in period $t$ |
| $C_t^m$ | Unit cost of labor in period $t$ |
| $C_t^c$ | cost of hiring an employee in period $t$ |
| $C_t^D$ | cost of dismissing an employee in period $t$ |
| $C_t^S$ | Subcontracting cost in period $t$ |
| $C_t^f$ | Out-of-stock cost in the period $t$ |
| $C_t^{He}$ | Overtime cost in period $t$ |
| **DECISION VARIABLES** | |
| $U_t$ | number of units produced in period $t$ |
| $U_t^S$ | number of units produced under subcontracting in period $t$ |
| $U_t^e$ | number of units in stock at the end of period $t$ |
| $U_t^f$ | number of units missing in the period $t$ |
| $T_t$ | number of workers available in the period $t$ |
| $T_t^c$ | number of workers hired in period $t$ |
| $T_t^D$ | number of workers laid off in period $t$ |
| $Ca_t$ | Production production capacity in period $t$ |
| $D_t^e$ | Contracted power demand in period $t$ |
| $E_f$ | inventory at the end of the total period |
| $H_t^n$ | number of normal hours worked in period $t$ |
| $H_t^e$ | number of overtime hours worked in period $t$ |

### Appendix A

**Table A1.** Power demand consumption history.

| Month | Potency (kw) | Month | Potency (kw) | Month | Potency (kw) | Month | Potency (kw) | Month | Potency (kw) |
|---|---|---|---|---|---|---|---|---|---|
| Jan/2014 | 204 | Jan/2015 | 240 | Jan/2016 | 252 | Jan/2017 | 259 | Jan/2018 | 221 |
| Feb/2014 | 216 | Feb/2015 | 252 | Feb/2016 | 252 | Feb/2017 | 245 | Feb/2018 | 201 |
| Mar/2014 | 264 | Mar/2015 | 240 | Mar/2016 | 252 | Mar/2017 | 235 | Mar/2018 | 211 |
| Apr/2014 | 240 | Apr/2015 | 240 | Apr/2016 | 252 | Apr/2017 | 245 | Apr/2018 | 202 |
| May/2014 | 216 | May/2015 | 252 | May/2016 | 264 | May/2017 | 250 | May/2018 | 202 |

**Table A1.** *Cont.*

| Month | Potency (kw) | Month | Potency (kw) | Month | Potency (kw) | Month | Potency (kw) | Month | Potency (kw) |
|---|---|---|---|---|---|---|---|---|---|
| Jun/2014 | 216 | Jun/2015 | 264 | Jun/2016 | 252 | Jun/2017 | 264 | Jun/2018 | 206 |
| Jul/2014 | 228 | Jul/2015 | 252 | Jul/2016 | 252 | Jul/2017 | 250 | Jul/2018 | 216 |
| Aug/2014 | 252 | Aug/2015 | 240 | Aug/2016 | 250 | Aug/2017 | 259 | Aug/2018 | 226 |
| Sep/2014 | 252 | Sep/2015 | 240 | Sep/2016 | 235 | Sep/2017 | 264 | Sep/2018 | 221 |
| Oct/2014 | 252 | Oct/2015 | 264 | Oct/2016 | 245 | Oct/2017 | 259 | Oct/2018 | 206 |
| Nov/2014 | 240 | Nov/2015 | 276 | Nov/2016 | 259 | Nov/2017 | 254 | Nov/2018 | 216 |
| Dec/2014 | 288 | Dec/2015 | 252 | Dec/2016 | 254 | Dec/2017 | 226 | Dec/2018 | 221 |

**Table A2.** Electricity consumption history.

| Month | Energy (kwh) | Month | Energy (kwh) | Month | Energy (kwh) | Month | Energy (kwh) | Month | Energy (kwh) |
|---|---|---|---|---|---|---|---|---|---|
| Jan/2014 | 66,203 | Jan/2015 | 83,532 | Jan/2016 | 70,272 | Jan/2017 | 86,027 | Jan/2018 | 79,529 |
| Feb/2014 | 75,792 | Feb/2015 | 100,980 | Feb/2016 | 89,592 | Feb/2017 | 82,935 | Feb/2018 | 34,769 |
| Mar/2014 | 59,220 | Mar/2015 | 89,136 | Mar/2016 | 88,008 | Mar/2017 | 68,050 | Mar/2018 | 69,184 |
| Apr/2014 | 74,364 | Apr/2015 | 78,108 | Apr/2016 | 85,044 | Apr/2017 | 82,784 | Apr/2018 | 76,301 |
| May/2014 | 59,832 | May/2015 | 75,420 | May/2016 | 87,216 | May/2017 | 75,050 | May/2018 | 71,020 |
| Jun/2014 | 65,304 | Jun/2015 | 92,784 | Jun/2016 | 87,636 | Jun/2017 | 89,838 | Jun/2018 | 70,393 |
| Jul/2014 | 94,164 | Jul/2015 | 91,308 | Jul/2016 | 53,832 | Jul/2017 | 93,682 | Jul/2018 | 79,856 |
| Aug/2014 | 102,504 | Aug/2015 | 84,912 | Aug/2016 | 89,736 | Aug/2017 | 102,359 | Aug/2018 | 73,906 |
| Sep/2014 | 99,948 | Sep/2015 | 70,188 | Sep/2016 | 66,703 | Sep/2017 | 102,745 | Sep/2018 | 73,524 |
| Oct/2014 | 52,356 | Oct/2015 | 95,364 | Oct/2016 | 77,791 | Oct/2017 | 69,419 | Oct/2018 | 72,326 |
| Nov/2014 | 84,516 | Nov/2015 | 81,852 | Nov/2016 | 95,322 | Nov/2017 | 88,272 | Nov/2018 | 72,262 |
| Dec/2014 | 107,928 | Dec/2015 | 96,444 | Dec/2016 | 88,956 | Dec/2017 | 80,510 | Dec/2018 | 72,413 |

**Table A3.** Potato production history.

| Month | Production (kg) | Month | Production (kg) | Month | Production (kg) | Month | Production (kg) | Month | Production (kg) |
|---|---|---|---|---|---|---|---|---|---|
| Jan/2014 | 71,449.80 | Jan/2015 | 163,406.92 | Jan/2016 | 94,561.22 | Jan/2017 | 108,315.34 | Jan/2018 | 52,178.56 |
| Feb/2014 | 98,108.06 | Feb/2015 | 122,309.88 | Feb/2016 | 102,871.56 | Feb/2017 | 67,618.20 | Feb/2018 | 15,807.6 |
| Mar/2014 | 75,186.06 | Mar/2015 | 160,586.38 | Mar/2016 | 145,226.16 | Mar/2017 | 85,486.90 | Mar/2018 | 90,957.72 |
| Apr/2014 | 89,184.04 | Apr/2015 | 101,419.64 | Apr/2016 | 112,505.82 | Apr/2017 | 91,313.48 | Apr/2018 | 94,088.215 |
| May/2014 | 51,441.18 | May/2015 | 97,440.4 | May/2016 | 120,193.48 | May/2017 | 91,153.62 | May/2018 | 62,223.9 |
| Jun/2014 | 73,104.84 | Jun/2015 | 110,822.38 | Jun/2016 | 104,055.4 | Jun/2017 | 95,449.86 | Jun/2018 | 99,377.09 |
| Jul/2014 | 172,998.24 | Jul/2015 | 123,069.82 | Jul/2016 | 50,575.16 | Jul/2017 | 130,865.92 | Jul/2018 | 95,340.575 |
| Aug/2014 | 170,871.62 | Aug/2015 | 99,924.98 | Aug/2016 | 132,796.44 | Aug/2017 | 154,074.48 | Aug/2018 | 83,872.86 |
| Sep/2014 | 195,048.08 | Sep/2015 | 77,072.10 | Sep/2016 | 84,820.46 | Sep/2017 | 136,504.22 | Sep/2018 | 75,521.85 |
| Oct/2014 | 98,406.44 | Oct/2015 | 127,472.06 | Oct/2016 | 98,265.96 | Oct/2017 | 97,858.3 | Oct/2018 | 78,968.54 |
| Nov/2014 | 15,5987.16 | Nov/2015 | 100,275.86 | Nov/2016 | 121,060.3 | Nov/2017 | 112,327.76 | Nov/2018 | 78,199.93 |
| Dec/2014 | 185,057.78 | Dec/2015 | 127,309.30 | Dec/2016 | 124,903.44 | Dec/2017 | 98,389.48 | Dec/2018 | 96,792.86 |

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
