# Peer review of "Model for Integrating the Electricity Cost Consumption and Power Demand into Aggregate Production Planning"

_applsci, doi:10.3390/app12157577_

Round 1

Reviewer 1 Report

The article develops a model of aggregate production planning by integrating the electricity cost consumption and power demand. Some minor comments are:

1   1. Introduction

It is a bit weird how the order of referenced literature appear in the article. It starts with “[41]”.

2   2. Materials and Methods

The description of four steps in text (page2 line 76 to 78) does not match contents in Figure 1.

Page 3, line 89, please clarify the type of contract (blue or green modes?)

3   3. Results

It would make more sense to separate the description of “data collected” part from results section.

Page 8 line 312, did not see the table in appendix A.

Figure 7, the label for x-axil and y-axial is a bit unclear, please improve the resolution.  

Add units for Table 4, 5 and 6.

Appendix A

Suggest changing “Appendix A” to “Nomenclature” and place it in an appropriate location, since it is just a clarification of all the variables.

Author Response

Reviewer 1: The article develops a model of aggregate production planning by integrating the electricity cost consumption and power demand. Some minor comments are:  1. Introduction (It is a bit weird how the order of referenced literature appear in the article. It starts with “[41]”.)

Answer: We fully appreciate the suggestion and we adequate according.

Reviewer 1: Materials and Methods (The description of four steps in text (page2 line 76 to 78) does not match contents in Figure 1.)

Answer: The description was reordered as (i) Data collection; (ii) Power Demand Definition; (iii) Production Demand Forecast; and (iv) Aggregate Planning Modeling and Solution.

Reviewer 1: Materials and Methods (Page 3, line 89, please clarify the type of contract (blue or green modes?))

Answer: We appreciate the suggestion according the contract. The peak time, defined by ANEEL - National Electric Energy Agency - In its Resolution 414/2010, takes place between 6 pm and 9 pm when electricity consumption is higher, except on Saturdays, Sundays, and national holidays. The off-peak time is the period composed of the set of consecutive daily hours and complementary to those defined during peak hours.

To discourage the use of electricity at peak hours by high voltage consumers, higher tariffs are imposed at that time. For this reason there are two modalities in the electricity bill, blue and green. Consumers powered by 69kV or more are charged only in the blue mode, while the others can choose blue or green, whichever is more convenient for the consumer. In the blue mode, there is a tariff differentiated by the contracted power demand at the peak and off-peak, and in the green mode, the power demand tariff is unique, and can be consumed both at the peak and off [1].

Reviewer 1: Results (It would make more sense to separate the description of the "data collected" part from the results section.)

Answer: Accomplished. Getting Started 3.1 Data Collected.

Reviewer 1: Results (Page 8 line 312, did not see the table in appendix A.)

Answer: Inserted as suggested.

Reviewer 1: Results (Figure 7, the label for x-axis and y-axial is a bit unclear, please improve the resolution.)

Answer: Adjusted as suggested.

Reviewer 1: Results (Add units for Table 4, 5 and 6.)

Answer: This information was added in the full text.

Period(month), Contractors (number of people), Fired (number of people), Workforce (number of people), Production in overtime (hours), Production (kg), Power demand (kW), Production demand (kg).

Reviewer 1: Appendix A (Suggest changing “Appendix A” to “Nomenclature” and place it in an appropriate location, since it is just a clarification of all the variables.)

Answer: We fully appreciate the suggestion; we replaced in full text with the Nomenclature list.

Reviewer 2 Report

·     The references numbers start with [41] and continue not following a linear numeration. Please, numerate the references in the order in which these are cited in the paper for the first time.

·     There are many English language errors in the text. Here only some examples follow in order to guide the authors through a required language revision of the full text.

o   Page 1, line 18: “To” should be substituted with “into”, like in the title of the paper;

o   Page 1, line 22: in the phrase “Before the modeling, the new power demand was calculated and, finally, the verification of the model solution.” A verb is missing.

o   Page 1, line 24: “with electric” should be “of electric”

o   Page 2, line 54: “ensure supply to customers of energy supply networks”

o   Page 2, line 54-56: Hu et al. [22] used 54 forecasting models to predict the energy consumption of pipelines to meet their energy 55 needs optimizing management: please, insert some commas to improve the legibility of the phrase

o   Page 3, line 89: since in the rest of the phrase you used “;”, please put “:” after “collect”

o   Page 4, line 131: “The R Core Team [38] developed the R Cran software was used to read and model the demand data”. Maybe “that” is missing before “was used”

o   Page 17, line 538: “model” is repeated in the phrase

·     Page 1, line 42: please specify better what you mean with “due to the seasons”; is it something linked to the global warming or to some seasons that in Brazil are systematically characterized by a low amount of rainfalls?

·     Page 2, lines 76-78: the four steps described don’t match with the stages described in Figure 1 and in the following sub-chapters

·     Page 3, lines 89-90: please, define blue and green modes

·     Page 3, line 115-119, “the proposed model…”: it is not clear if the TCi and TUi are the ones defined by the law (TCi=TDPC and Tui=2xTDPC)

·     Page 6, line 262: “2,91 billion”, the currency is missing.

·     Page 7, line 293, “must pay double”: it should be better to insert “for the exceeding part”.

·     Page 7, chapter 3.2: you used often “in the last five years” but the cited data refer to the period 2014-2018 and now we are in year 2022.

·     Page 7, lines 295-296 and Page 8, line 312: you state that the data on the electric power consumption is available in Appendix A, but, if you look at the latter (pages 18-19), no data can be found

·     Page 8, line 312: please verify which equation has been actually employed to plot the graph in Figure 4.

·     Page 9, line 337: please specify better what “and the lag’s horizontal axis” means

·     Page 10, Figure 7: the differentiating process has been introduced but not described in detail. Please provide an explanation on how, from data, you plot seasonal, trend and remainder.

·     From Chapter 3.5 on: it is not clear if and/or how the data present in Table 2 (production demand forecast) have been used in the further developments

Author Response

Reviewer 2: The references numbers start with [41] and continue not following a linear numeration. Please, enumerate the references in the order in which these are cited in the paper for the first time.

Answer: Inserted as suggested.

Reviewer 2: There are many English language errors in the text. Here only some examples follow in order to guide the authors through a required language revision of the full text.

Answer: We fully appreciate the suggestion; a native speaker and Grammarly Premium revised the manuscript.

Reviewer 2: Page 1, line 18: “To” should be substituted with “into”, like in the title of the paper;

Answer: Was changed to "power demand into the aggregate production planning.”

Reviewer 2:  Page 1, line 22: in the phrase “Before the modeling, the new power demand was calculated and, finally, the verification of the model solution.” A verb is missing.

Answer: Before modeling, the new energy demand was calculated, and, finally, the model solution verification was performed.

Reviewer 2: Page 1, line 24: “with electric” should be “of electric”

Answer: Changed as suggested.

Reviewer 2:  Page 2, line 54: “ensure supply to customers of energy supply networks”

Answer: Was changed to "Ensure the supply to customers of energy supply networks."

Reviewer 2:  Page 2, line 54-56: Hu et al. [22] used 54 forecasting models to predict the energy consumption of pipelines to meet their energy 55 needs optimizing management: please, insert some commas to improve the legibility of the phrase

Answer: We appreciate the suggestion, and the phrase was rewritten.

Reviewer 2: Page 3, line 89: since in the rest of the phrase you used “;”, please put “:” after “collect”

Answer: Changed as suggested.

Reviewer 2: Page 4, line 131: “The R Core Team [38] developed the R Cran software was used to read and model the demand data”. Maybe “that” is missing before “was used”

Answer: We fully appreciate the suggestion and was changed as "The R Core Team [38] developed the R Cran software that was used to read and model the demand data. The TSA, MASS, tseries, and forecast packages were used for the modeling."

Reviewer 2: Page 17, line 538: “model” is repeated in the phrase

Answer: This presented model was formulated with linear programming, following Rajaram and Karmarkar [45], who state that APP models are generally designed as linear programming that minimizes costs.

Reviewer 2: Page 1, line 42: please specify better what you mean with “due to the seasons”; is it something linked to the global warming or to some seasons that in Brazil are systematically characterized by a low amount of rainfalls?

Answer: In the country, some seasons are systematically characterized by a low amount of rainfall, which influences water resources [10, 20]. 

Reviewer 2: Page 2, lines 76-78: the four steps described don’t match with the stages described in Figure 1 and in the following sub-chapters

Answer: (i) Data collection; (ii) Power Demand Definition; (iii) Production Demand Forecast; and (iv) Aggregate Planning Modeling and Solution.

Reviewer 2: Page 3, lines 89-90: please, define blue and green modes

Answer: The peak time, defined by ANEEL - National Electric Energy Agency - In its Resolution 414/2010, occurs between 6 pm and 9 pm when electricity consumption is higher, except on Saturdays, Sundays, and national holidays. The off-peak time is the period composed of the set of consecutive daily hours and complementary to those defined during peak hours.

To discourage the use of electricity at peak hours by high voltage consumers, higher tariffs are imposed at that time. For this reason, the electricity bill has two modalities: blue and green. Consumers powered by 69kV or more are charged only in the blue mode, while the others can choose blue or green, whichever is more convenient for the consumer. In the blue mode, there is a tariff differentiated by the contracted power demand at the peak and off-peak, and in the green mode, the power demand tariff is unique and can be consumed both at the peak and off [1].

Reviewer 2: Page 3, line 115-119, “the proposed model…”: it is not clear if the TCi and TUi are the ones defined by the law (TCi=TDPC and Tui=2xTDPC)

Answer: TCi and TUi ANEEL defines them.

Reviewer 2: Page 6, line 262: “2,91 billion”, the currency is missing.

Answer:  Line 278 of 2.91 billion reais

Reviewer 2: Page 7, line 293, “must pay double”: it should be better to insert “for the exceeding part”.

Answer: The power demand contracted by the industry in the last contract period was 255 kW. If more than 5% of the contracted demand is consumed, the consumer unit for pay the exceeding part.

Reviewer 2: Page 7, chapter 3.2: you used often “in the last five years” but the cited data refer to the period 2014-2018 and now we are in year 2022.

Answer: This represents the average value from January monthly indicators 2014 to December 2018. From the data collected, all the power demands measured from January 2014 to December 2018 were organized in a Table (Nomenclature).

Reviewer 2: Page 7, lines 295-296 and Page 8, line 312: you state that the data on the electric power consumption is available in Appendix A, but, if you look at the latter (pages 18-19), no data can be found

Answer: Inserted as suggested.

Reviewer 2: Page 8, line 312: please verify which equation has been actually employed to plot the graph in Figure 4.

Answer: Done, equation 3.

Reviewer 2: Page 9, line 337: please specify better what “and the lag’s horizontal axis” means

Answer: In Figure 6, the vertical axis indicates autocorrelation, and the horizontal axis indicates lag. The blue dashed line indicates where it is significantly different from zero. Note that all values except the first are within the blue dashed line limit. This means zero auto-correlation, indicating that potato production can be random and supposedly stationary. If the values were above the dashed line, the series would not be random and would have to be treated with a moving average, which is not the case. However, to be sure of the statements, it was necessary to carry out a test to prove the assumptions.

Reviewer 2: Page 10, Figure 7: the differentiating process has been introduced but not described in detail. Please explain how, from the data, you plot seasonal, trend, and remainder.

Answer: The STL algorithm performs time series smoothing using the LOESS algorithm [37] in two loops; the inner loop iterates between seasonal and trend smoothing, and the outer loop minimizes the effect of outliers. During the inner loop, the seasonal component is calculated first and removed to calculate the trend component. The remainder is calculated by subtracting the seasonal and trend components from the time series. Decomposing seasonal trends using LOESS (STL) is a robust time series decomposition method often used in economic and environmental analyses. The STL method uses locally adjusted regression models to decompose a time series into trend, seasonal and remainder components [38-39-40].

Reviewer 2: From Chapter 3.5, it is unclear if and how the data present in Table 2 (production demand forecast) have been used in further developments.

Answer: The demand forecast performed will be used in aggregate production planning. Because, to do it, it is necessary to know demand forecast.

4.3 - To perform aggregate planning, you need demand forecast data. As the studied company did not have its demand forecast, it was necessary to perform it. Otherwise, only data would be collected.

Reviewer 3 Report

1. The abstract section should be more intensively focused on the main idea directly and must contain the contribution of this manuscript with numerical result indicators.

2. Motivations for conducting this study are not getting clear in the introduction.

3. The research question is not clearly outlined in this paper.

4. Problem with the reference number in the introduction section.

5. A separate section for the review of literature is recommended to be expounded. Add more ideas from additional literature and studies.

6. More state-of-the-art literature review should be undertaken to cover various applications of the proposed approach.

7. More explication of the power demand to be contracted (DEC).

8. All equations must be clearly referenced.

9. Please have an introductory paragraph on the results and discussion.

10. The figures are not clearly presented. Poor qualities of figures, please enhance the resolution of all figures.

11. Some sentences may have grammatical errors or are hard to follow. Please check the manuscript properly and proofread it.

12. The conclusion section should be rearranged, and numerical results should be added. Also, the authors may propose some interesting problems as future work in the conclusion.

Author Response

Reviewer 3: The abstract section should be more intensively focused on the main idea directly and must contain the contribution of this manuscript with numerical result indicators.

Answer: We fully appreciate the suggestions; the abstract was revised according to your suggestion.

Reviewer 3: Motivations for conducting this study are not getting clear in the introduction.

Answer: We appreciate the suggestion, and a paragraph was added in the introduction section.

Reviewer 3: The research question is not clearly outlined in this paper.

Answer: We fully appreciate the suggestion, and the research question was inserted in the manuscript.

Reviewer 3: Problem with the reference number in the introduction section.

Answer: We fully appreciate the suggestion, and the reference number was adjusted in the manuscript.

Reviewer 3: A separate section for the review of literature is recommended to be expounded. Add more ideas from additional literature and studies.

Answer: We fully appreciate the suggestion, a section called theoretical background was inserted in the manuscript to fill this problem.

Reviewer 3: More state-of-the-art literature review should be undertaken to cover various applications of the proposed approach.

Answer: The literature was improved by the theoretical background section.

Reviewer 3: More explication of the power demand to be contracted (DEC).

Answer: The power demand to be contracted is mandatory and continuously made available by the distributor, at the point of delivery, according to the value and period of validity previously established in the contract, and which must be paid in full, whether or not used during the billing period, expressed in kilowatts (kW). During the entire contract period, for example, one year, the company pays monthly for the previously contracted power demand, even if it is not used. Overshoot occurs when the electric power utility measures a value above the power demand contracted by the unit. The rate for exceeding demand stipulated by ANEEL is double the contracted rate. In this condition, the company pays the contracted value plus the value of exceeding demand. In practice, companies normally contract the highest value of power demand measured between the months of the previous year to avoid exceeding demand during the next contract period.

Reviewer 3: All equations must be clearly referenced.

Answer: We intend to adjust as suggested.

Reviewer 3: Please have an introductory paragraph on the results and discussion.

Answer: We appreciate the suggestion, as the results section has six subsections; these were described in the first paragraph.

Reviewer 3: The figures are not clearly presented. Poor qualities of figures, please enhance the resolution of all figures.

Answer: We change as suggested.

Reviewer 3: Some sentences may have grammatical errors or are hard to follow. Please check the manuscript properly and proofread it.

Answer: The manuscript was revised by a native English speaker and Grammarly Premium.

Reviewer 3: The conclusion section should be rearranged, and numerical results should be added. Also, the authors may propose some interesting problems as future work in the conclusion.

Answer: We fully appreciate the suggestion, and we inserted in the conclusion section the information.

Round 2

Reviewer 2 Report

The authors have implemented all of the corrections required.

Reviewer 3 Report

No comments in this version.